# Reconstructing cell cycle and disease progression using deep learning

Philipp Eulenberg[1,2], Niklas Köhler[1,2], Thomas Blasi[1,3], Andrew Filby[4], Anne E. Carpenter [3], Paul Rees[3,5], Fabian J. Theis [1,6] & F. Alexander Wolf [1]

We show that deep convolutional neural networks combined with nonlinear dimension reduction enable reconstructing biological processes based on raw image data. We demonstrate this by reconstructing the cell cycle of Jurkat cells and disease progression in diabetic retinopathy. In further analysis of Jurkat cells, we detect and separate a sub-population of dead cells in an unsupervised manner and, in classifying discrete cell cycle stages, we reach a sixfold reduction in error rate compared to a recent approach based on boosting on image features. In contrast to previous methods, deep learning based predictions are fast enough for on-the-fly analysis in an imaging flow cytometer.

[1] Helmholtz Zentrum München—German Research Center for Environmental Health, Institute of Computational Biology, Neuherberg, Munich, Germany. [2] Department of Physics, Arnold Sommerfeld Center for Theoretical Physics, LMU München, Munich, Germany. [3] Imaging Platform at the Broad Institute of Harvard and Massachusetts Institute of Technology, Cambridge, MA, USA. [4] Flow Cytometry Core Facility, Faculty of Medical Sciences, Newcastle University, Newcastle upon Tyne, UK. [5] College of Engineering, Swansea University, Singleton Park, Swansea, UK. [6] Department of Mathematics, TU München, Munich, Germany. Philipp Eulenberg and Niklas Köhler contributed equally to this work. Correspondence and requests for materials should be addressed to F.J.T. (email: fabian.theis@helmholtz-muenchen.de) or to F.A.W. (email: alex.wolf@helmholtz-muenchen.de)

A major challenge and opportunity in biology is interpreting the increasing amount of information-rich and high-throughput single-cell data. Here, we focus on imaging data from fluorescence microscopy[1], in particular from imaging flow cytometry (IFC), which combines the fluorescence sensitivity and high-throughput capabilities of flow cytometry with single-cell imaging[2]. Imaging flow cytometry is unusually well-suited to deep learning as it provides very high sample numbers and image data from several channels, that is, high dimensional, spatially correlated data. Deep learning is therefore capable of processing the dramatic increase in information content—compared to spatially integrated fluorescence intensity measurements as in conventional flow cytometry[3]—in IFC data. Also, IFC provides one image for each single cell, and hence does not require whole-image segmentation.

Deep learning enables improved data analysis for high-throughput microscopy as compared to traditional machine learning methods[4–7]. This is mainly due to three general advantages of deep learning over traditional machine learning: there is no need for cumbersome preprocessing and manual feature definition, prediction accuracy is improved, and learned features can be visualized to uncover their biological meaning. In particular, we demonstrate that this enables reconstructing continuous biological processes, which has stimulated much research effort in the past years[8–11]. Only one of the other recent works on deep learning in high-throughput microscopy discusses the visualization of network features[12], but none deal with continuous biological processes[12–16].

When aiming at an understanding of a specific biological process, one often only has coarse-grained labels for a few qualitative stages, for instance, cell cycle or disease stages. While a continuous label could be efficiently used in a regression based approach, qualitative labels are better used in a classification-based approach. In particular, if the ordering of the categorical labels at hand is not known, a regression based approach will fail. Also, the detailed quantitative information necessary for a continuous label is usually only available if a phenomenon is already understood on a molecular level and markers that quantitatively characterize the phenomenon are available. While this is possible for cell cycle when carrying out elaborate experiments where such markers are measured[5, 8], in many other cases, this is too tedious, has severe side effects with unwanted influences on the phenomenon itself or is simply not possible as markers for a specific phenomenon are not known. Therefore, we propose a general workflow that uses a deep convolutional neural network combined with classification and visualization based on nonlinear dimension reduction (Fig. 1).

## Results

**Reconstructing cell cycle progression.** To show how learned features of the neural network can be used to visualize, organize, and biologically interpret single-cell data, we study the activations in the last layer of the neural network[17]. The approach is motivated by the fact that the neural network strives to organize data in the last layer in a linearly separable way, given that it is directly followed by a softmax classifier. Distances from the separating hyperplanes in this space can be interpreted as similarities between cells in terms of the features extracted by the network. Cells with similar feature representations are close to each other and cells with different class assignments are far away from each other. This gives a much more fine-grained notion of biological similarity than provided by the class labels used for labeling the training set. Evidently, it automatically generalizes to the unseen, new data in the validation data set. The activation space of our network's last layer is much too high dimensional to be accessible for human interpretation. We use nonlinear dimension reduction to visualize the data in a lower dimensional space, in particular, t-distributed stochastic neighbor embedding (tSNE)[18].

We apply the approach to raw IFC images of 32,266 asynchronously growing immortalized human T-lymphocyte cells (Jurkat cells)[5, 19], which can be classified into seven different stages of cell cycle (Fig. 2), including phases of interphase (G1, S, and G2) and phases of mitosis (Prophase, Anaphase, Metaphase,

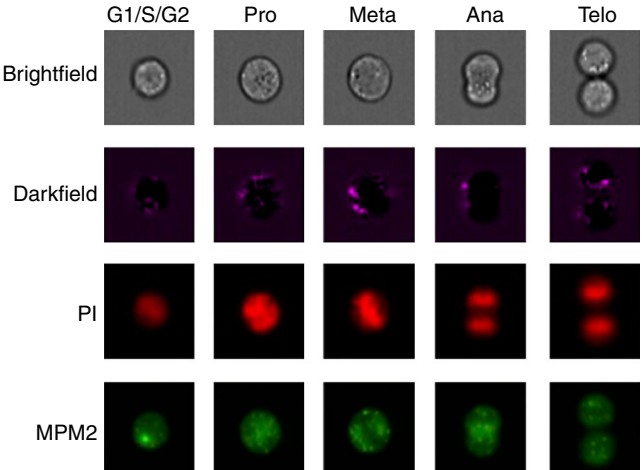

**Fig. 2** Representative images for the cell cycle stages as measured in *brightfield*, *darkfield*, and *fluorescence channels*. Seven cell cycle stages define seven classes. We only show one representative image for the interphase classes G1, S, and G2, which can hardly be distinguished by eye

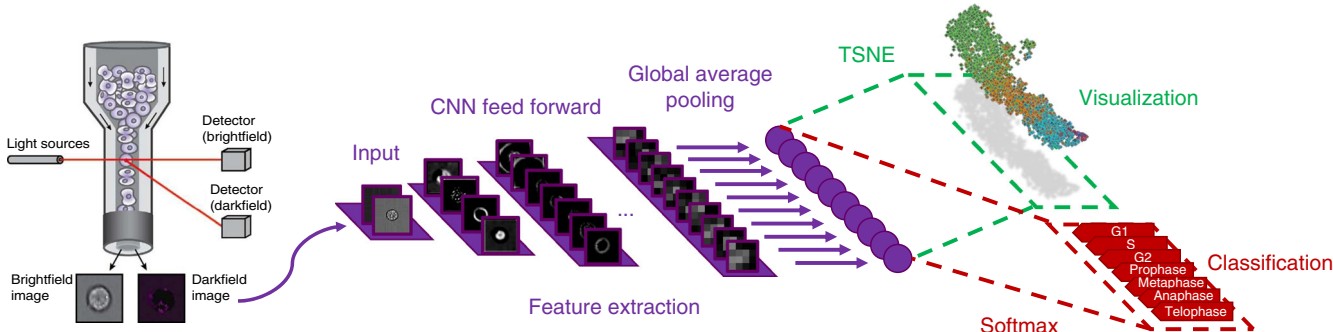

**Fig. 1** Overview of analysis workflow. Images from all channels of a high-throughput microscope are uniformly resized and directly fed into the neural network, which is trained using categorical labels. The learned features are used for both classification and visualization

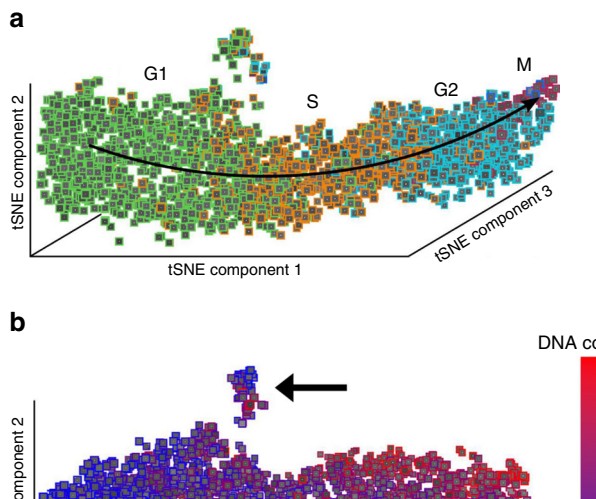

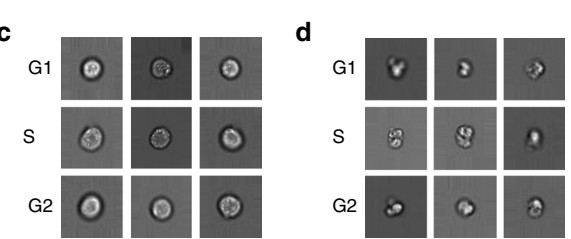

**Fig. 3** Cell-cycle reconstruction and detection of abnormal cells. **a** tSNE visualization of the validation data set in activation space representation. All interphase classes (G1, S, G2) and the two mitotic phases with the highest number of representatives are shown (Prophase: *red*, Metaphase: *blue*). Telophase and Anaphase are not visible due to their low number representatives. **b** tSNE visualization of data from the interphase classes (G1, S, G2) in activation space. The *color map* now shows the DNA content of cells. A cluster of damaged cells is indicated with an arrow. **c** Randomly picked representatives from the bulk of undamaged cells. **d** Randomly picked representatives from the cluster of damaged cells

and Telophase). We observe that the Jurkat cell data is organized in a long stretched cylinder along which cell cycle phases are ordered in the chronologically correct order (Fig. 3a and Supplementary Movie 1). This is remarkable as the network has been provided with neither structure within the class labels nor the relation among classes. The learned features evidently allow reconstructing the continuous temporal progression from the raw IFC data, and by that define a continuous distance between the phenotypes of different cell cycle phases.

We separately visualized just those cells annotated as being in the interphase classes (G1, S, G2) (Fig. 3b) and colored them with the DNA content obtained from one of the fluorescent channels of the IFC. The DNA content reflects the continuous progression of cells in G1, S, and G2 on a more fine-grained level. Its correspondence with the longitudinal direction of the cylinder found by tSNE demonstrates that the temporal order learned by the neural network is accurate even beyond the categorical class labels.

**Detecting abnormal cells in an unsupervised manner.** Both tSNE visualizations (Fig. 3a, b) produce a small, separate cluster highlighted with an arrow in Fig. 3b. This cluster is learned in an unsupervised way as cell cycle phase labels provide no

information about it: it contains cells from all three interphase classes. While cells in the bulk have high circularity and well defined borders (Fig. 3c), cells in the small cluster are characterized by morphological abnormalities such as broken cell walls and outgrowths, signifying dead cells (Fig. 3d).

**Deep learning automatically performs segmentation.** We interpret the data representation encoded in one of the trained intermediate layers of the neural network by inspecting its activation patterns using exemplary input data from several classes (Fig. 4). These activation patterns are the essential information transmitted through the network. They show the response of various kernels on their input. By inspecting the activation patterns, we obtain an insight into what the network is "focusing on" in order to organize data. We observe a strong response to features that arise from the cell border thickness (Fig. 4, map 1), to area-based features (Fig. 4, map 2), as well as cross-channel features. For example, map 4 in Fig. 4 shows high response to the difference of information from the brightfield channel, as seen in map 2, and scatter intensities, as seen in map 3. A strong response of the neural network to area-based features as in map 2 could indicate that the network learned to perform a segmentation task.

**Deep learning outperforms boosting for cell classification.** We study the classification performance of deep learning on the validation data set shown in Fig. 3. We first focus on the case in which G1, S, and G2 phases are considered as a single class. Using fivefold cross-validation on the 32,266 cells, we obtain an accuracy of 98.73% ± 0.16%. This means a sixfold improvement in error rate over the 92.35% accuracy for the same task on the same data in prior work using boosting on features extracted via image analysis[5]. The confusion matrix obtained using boosting show high true positive rates for the mitotic phases (Fig. 5a). For example, no cells in Anaphase and Telophase are wrongly classified, as indicated by the zeros in the off-diagonal entries of the two lower rows of the matrix (Fig. 5a). This means high sensitivity, most cells from mitotic phases are correctly classified as such. Still this comes at the price of low precision: many cells from the interphase class are classified as mitotic phases, as indicated by the high numbers in the off-diagonal entries of the first row of the matrix (Fig. 5a). Deep learning, by contrast, achieves high sensitivity and precision, leading to an almost diagonal confusion matrix (Fig. 5b). Further deep learning allows to classify all seven cell cycle stages with an accuracy of 79.40% ± 0.77% (Supplementary Notes and Supplementary Fig. 2).

**Reconstructing disease progression.** To substantiate the generality of our results, consider now a data set that is related to diabetic retinopathy, which is the leading cause of blindness in the working-age population of the developed world. We study 30,000 color fundus photographies of the human retina, which were classified into four disease states "healthy", "mild", "medium", and "severe". We observe a reconstructed disease progression (Fig. 6) for 8000 samples in the validation data set, that is, the four disease states are ordered along disease severity, even though the network has not been provided with the ordering information. Similar to the cell cycle example, the ordering ensures that only neighboring classes overlap, as visible from the tSNE plot (Fig. 6a).

**Discussion**

The visualization of the data as encoded in the last layer of the network using tSNE demonstrates how deep learning overcomes a well known issue of traditional machine learning. When trained

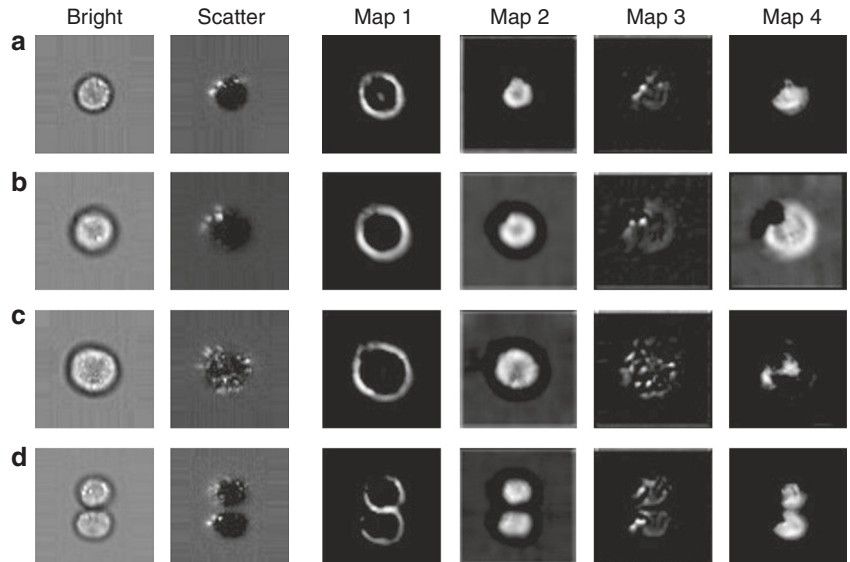

**Fig. 4** Exemplary activation patterns of intermediate layers. Plotted are activations after the second convolutional module for examples of single cells from four different phases: **a** G1, **b** G2, **c** Anaphase, and **d** Telophase. The response maps mark regions of high activation. Map 1 responds to the cell boundaries. Map 2 responds to the internal area of the cells. Map 3 extracts the localized scatter intensities. Map 4 constitutes a cross-channel feature, which correlates with the difference of map 2 and 3

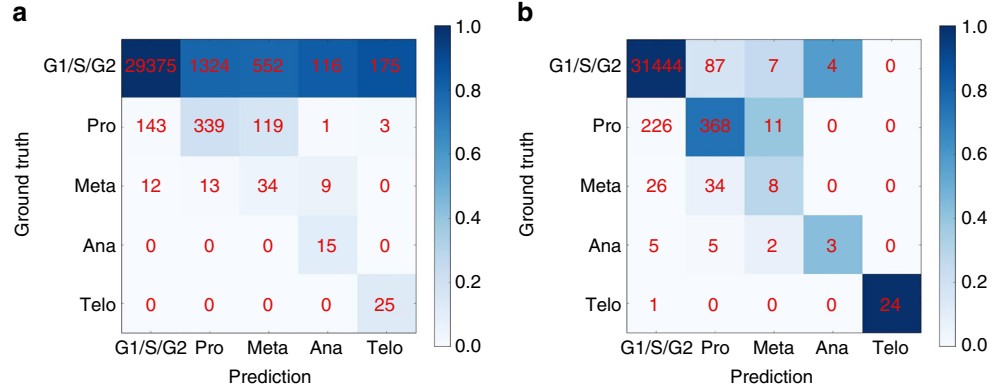

**Fig. 5** Confusion matrices for boosting and deep learning for classification of five classes. To compare with previous work[5], the three interphase phases (G1, S, G2) are treated as a single class. *Red* numbers denote absolute numbers of cells in each entry of the confusion matrix, that is, diagonal elements correspond to precision. Coloring of the matrix is obtained by normalizing absolute numbers to column sums. **a** Boosting[5], which leads to 92.35% accuracy. **b** Deep learning, which leads to 98.73% ± 0.16% accuracy

on a continuous biological process using discrete class labels, traditional machine learning often fails to resolve the continuum[4]. Reconstructing continuous biological processes though is possible in the context of so-called pseudotime algorithms[9–11]. For the cell cycle it has been demonstrated[8], but in a very different setting. These authors measured five stains that uniquely define the cell cycle and then applied a pseudotime algorithm[9] within this five-dimensional space. This procedure is only possible if stains that correlate with a given process of interest are known, if they do not interact with the process and if the elaborate experiments for measuring the intensity of these stains can be carried out. We, by contrast, use raw images directly and the learned features of the neural network automatically constitute a feature space in which data is continuously organized. In the Supplementary Notes, we demonstrate that pseudotime algorithms fail at solving this much harder problem.

Deep learning is able to reconstruct continuous processes based on categorical labels as adjacent classes are morphologically more similar than classes that are temporally further separated. If this assumption does not hold, also pseudotime algorithms fail to reconstruct a process. This can be better understood when inspecting Fig. 6a, where we show the tSNE visualization of the validation set for the diabetic retinopathy (DR) data. Samples are organized in the correct order of progression through disease states, from healthy to severe DR. However, between the healthy cluster (*green*) and the mild DR cluster (*orange*), one observes an area of slightly reduced sampling density (*dashed line*). This should not be attributed to "less data points having been sampled in this region" but should be seen as a consequence of the fact that the overlap between the "healthy" stage and the "mild" stage is smaller than the overlap of the diseased stages among each other. If there was no overlap between "healthy" and "mild" stages, the tSNE would show a complete separation of the healthy cluster from the rest of the data. Such a behavior is typically observed if the underlying data is not sampled from a continuous process.

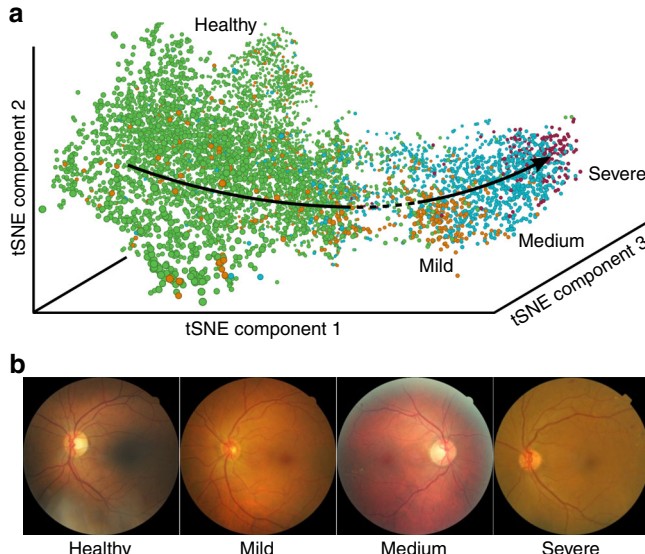

**Fig. 6** Reconstruction of disease progression in diabetic retinopathy. **a** tSNE visualization of activation space representation, colored according to the disease states. **b** Randomly chosen images for each class

The unsupervised detection of a discrete cluster of abnormal cells for the Jurkat cell data indicates that the neural network learns the cluster of abnormal cells independently of the cell-cycle-label based training. The model is therefore not only capable of resolving a biological process, but generates features that are general enough to separate incorrectly labeled cells that do not belong to the process. None of the mentioned pseudotime algorithms is capable of this. This shows the ability of deep learning to find unknown phenotypes and processes without knowledge about features or labels. Also, there is a high practical use of the detection of damaged cells. The Jurkat cell data set has been preprocessed using the IDEAS analysis software to remove images of abnormal cells. In particular, out of focus cells were removed by gating for images with gradient RMS and debris was removed by gating for circular objects with a large area. The discovery of a cluster of abnormal cells shows the limitations of this approach and provides a solution to it.

An advantage of using a neural network for cell classification in IFC is its speed. Traditional techniques rely on image segmentation and measurement, time-consuming processes limited to roughly 10 cells per second. Neural network predictions, by contrast, are extremely fast, as the main computation consists in parallelizable matrix multiplications ("forward propagations"), which can be performed using optimized numeric libraries. This yields a roughly 100-fold improvement in speed to about 1000 cells per second with a single GPU. Aside from much faster analysis of large cell populations, this opens the door to "sorting on-the-fly": imaging flow cytometers currently do not allow physically sorting individual cells into separate receptacles based on measured parameters, due to these speed limitations.

Given the compelling performance on reconstructing the cell cycle and diabetic retinophany, we expect deep learning to be helpful for understanding a wide variety of biological processes involving continuous morphology changes. Examples include developmental stages of organisms, dose response and the progression of healthy states to disease states, situations that have often been non-ideally reduced to binary classification problems. Ignoring intrinsic heterogeneity has likely hindered a deeper insight into the mechanisms at work. Analysis as demonstrated here could reveal morphological signatures at much earlier stages than previously recognized.

Our results indicate that reconstructing biological processes is possible for a wide variety of image data, if enough samples are available. Although generally lower-throughput in terms of the number of cells processed, conventional microscopy is nevertheless still high-throughput and can usually provide higher resolution images than IFC. Furthermore, given that multi-spectral methods are advancing rapidly, imaging mass spectrometry is allowing dozens of labeled channels to be acquired[20, 21]. Due to its basic structure and high flexibility, a deep learning framework like the one presented here can accommodate a large increase in the number of available channels.

We acknowledge discussions with the authors of related work, which became available as a preprint just before publication of the present paper[22].

## Methods

In the data set of 32,266 Jurkat cells, labeling is based on two fluorescent stains: propidium iodine (PI) to quantify each cell's DNA content and the mitotic protein monoclonal #2 (MPM2) antibody to identify cells in mitotic phases. These stains allow each cell to be labeled through a combination of algorithmic segmentation, morphology analysis of the fluorescence channels, and user inspection[5]. Note that 97.78% of samples in the data set belong to one of the interphase classes G1, S, and G2. The strong class imbalance in the data set is related to the fact that interphase lasts—when considering the actual length of the biological process—a much longer period of time than mitosis.

Recent advances in deep learning have shown that deep neural networks are able to learn powerful feature representations[23–26]. Based on the widely used "Inception" architecture[25], we developed the "DeepFlow" architecture, which is optimized for the relatively small input dimensions of IFC data. DeepFlow consists in 13 three-layer "dual-path" modules (Supplementary Fig. 3), which process and aggregate visual information at an increasing scale. These 39 layers are followed by a standard convolution layer, a fully connected layer and the softmax classifier. Training this 42-layer deep network does not present any computational difficulty, as the first three layers consist in reduction dual-path modules (Supplementary Fig. 3b), which strongly reduce the original input dimensions prior to convolutions in the following normal dual-path modules. The number of kernels used in each layer increases towards the end, until 336 feature maps with size $8 \times 8$ are obtained. A final average pooling operation melts the local resolution of these maps and generates the last 336-dimensional layer, which serves as an input for both classification and visualization.

This neural network operates directly on uniformly resized images. It is trained with labeled images using stochastic gradient descent with standard parameters (Supplementary Notes). For the IFC data, we focus on the case in which only brightfield and darkfield channels are used as input for the network, during training, visualization and prediction. As stated before, this case is of high interest as a fluorescent markers might affect the biological process under study or adequate markers are not known. We note, however, that technical imperfections in the IFC data capture might always lead to a minor amount of fluorescence signal, activated by a fluorescence channel, in the darkfield and brightfield channels, a phenomenon known as "bleed through" (Supplementary Notes).

**Code availability**. Code for the DeepFlow architecture and the Jurkat cell data set is available at https://github.com/theislab/deepflow.

**Data availability**. The retinopathy data set is available at https://www.kaggle.com/c/diabetic-retinopathy-detection/data and can be processed with standard packages and architectures.

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

## Acknowledgements

F.A.W. acknowledges support by the Helmholtz Postdoc Programme, Initiative and Networking Fund of the Helmholtz Association. P.R. and A.E.C. acknowledge the support of the Biotechnology and Biological Sciences Research Council/ National Science Foundation under grant BB/N005163/1 and NSF DBI 1458626. A.F. acknowledges support from the ISAC EL programme.

## Author contributions

P.E. and N.K. developed the deep learning model and the data analysis pipeline with equal contributions. F.J.T., T.B., and P.R. conceived the study. F.A.W. supervised the study with F.J.T. A.F. designed and performed the cell cycle experiments. F.A.W. performed the pseudotime-based reconstruction of cell cycle. F.A.W., N.K., P.E., and A.C. wrote the paper with help from all authors. All authors contributed to the interpretation of the results.

## Additional information

**Competing interests:** The authors declare no competing financial interests.

