## [Peer Review file · Nature Communications]

Reviewers' comments:

Reviewer #1 (Remarks to the Author):

This manuscript describes an application of deep learning techniques to an imaging flow cytometry dataset that focuses on the different stages of the cell cycle.

Major comments

The paper is mainly a technical application paper, where a classification approach is chosen to model the different stages of the cell cycle, and its main merit is an improvement of the accuracy of classifying the different cell cycle stages, compared to a previous approach where features were explicitly extracted using cell profiler and subsequently feed into a boosting type of classifier. However, the biological novelty is limited as no new biological insights regarding cell cycle progression are presented.

From the modelling perspective, the question can be raised whether the chosen classification approach is the best way to approach a problem that is essentially a "gradual" problem, and where regression-like methods, or the current novel class of trajectory inference/pseudo-temporal ordering methods would intuitively seem the better type of models to model such gradual transitioning phenomena. These models have recently been successfully applied to many single-cell technologies, including flow cytometry, mass cytometry and single-cell transcriptomics. As some of the authors of this paper have been involved in some of these methods, it is thus surprising that no comparison between the deep learning approach and pseudo-temporal ordering methods has been carried out

Detailed comments

* Figure 3a presents a 3D visualization of the cell "cycle", but this seems very linear. I wonder if this is an artifact of how the deep net was trained. Shouldn't we expect in theory that a cyclic structure would be the ideal lower-dimensional mapping of a cyclic event ?

* propodium iodine -> propidium iodine

* the provided movie contained an error and did not play

Reviewer #2 (Remarks to the Author):

The manuscript "Deep learning for imaging flow cytometry allows reconstructing cell cycle progression" by Eulenberg et al. describes a convolutional neural network combined with non-linear dimensionality reduction to analyze imaging flow cytometry data. As a proof of principle the authors apply the algorithms to images of Jurkat cells which were imaged using bright field and two cell cycle markers by imaging flow cytometry. Their computational approach then

reconstructed, without any supervision, a continuous representation of the cells along the cell cycle trajectory.

The manuscript is well written and the figures illustrate the analyses in a clear and concise manner. While the deep learning field is currently exploding and lots and lots of novel applications are emerging, there is still a lot to be done in the field of biological data analysis. As such the presented approach contains enough novelty to justify publication in Nature Communications. Also the algorithm will be useful to the scientific community. Before acceptance for publication, however, the authors need to address several points.

1. The authors adapt the Inception architecture and describe in their text the detailed structure of the network. Yet the motivation for the actual architecture is largely absent. Why 13 modules, why 326 feature maps with size 8x8 etc.? Please explain and ideally describe why these are the optimal numbers.

2. The algorithm was presented using imaging flow cytometry data. While their results are impressive, imaging mass cytometry is not too common. The manuscript would greatly increase in impact if the authors showed that their algorithm provides similar results using high content fluorescence imaging data. E.g. the authors could analyze the data from Gut et al. Nature Methods 2015. They already discuss this publication and in addition could also compare the performance of their algorithm directly to an existing method.

3. It is also not clear for which data DeepFlow recovers cellular continua. Is the presented example a "coincidence" or will DeepFlow always find a continuum given continuous data? Some synthetic data might be helpful here.

4. On page 6, I guess the reference goes to figure 4, not 6?

Reviewer 1

This manuscript describes an application of deep learning techniques to an imaging flow cytometry dataset that focuses on the different stages of the cell cycle.

Major comments

The paper is mainly a technical application paper, where a classification approach is chosen to model the different stages of the cell cycle, and its main merit is an improvement of the accuracy of classifying the different cell cycle stages, compared to a previous approach where features were explicitly extracted using cell profiler and subsequently feed into a boosting type of classifier. However, the biological novelty is limited as no new biological insights regarding cell cycle progression are presented.

From the modelling perspective, the question can be raised whether the chosen classification approach is the best way to approach a problem that is essential a "gradual" problem, and where regression-like methods, or the current novel class of trajectory inference/pseudo-temporal ordering methods would intuitively seem the better type of models to model such gradual transitioning phenomena. These models have recently been successfully applied to many single-cell technologies, including flow cytometry, mass cytometry and single-cell transcriptomics. As some of the authors of this paper have been involved in some of these methods, it is thus surprising that no comparison between the deep learning approach and pseudo-temporal ordering methods has been carried out

We thank the reviewer for her/his criticism of the manuscript. We though think that the main merit of the paper lies in showing for the first time that deep learning is much more powerful in reconstructing continuous processes than other known approaches.

Our previous submission did not contain a comparison to the mentioned pseudotime-inference and regression based methods as we deemed them not to be capable of solving the problem we were targeting: reconstructing a continuous process from categorical labels. This problem is highly relevant as often, one does not have "continuous labels" on which one could regress, but rather labels for a few qualitative stages of a process, as is, for instance, the case for cell cycle or disease stages. Detailed quantitative information is only available if the process is

already understood on a molecular level, for example, when markers are available that quantitatively characterize cell cycle or the progression of a disease. While this is possible for cell cycle when carrying out elaborate experiments where such markers are measured (Gut *et al.*, Nat. Meth., 2015), in many other cases, this is too tedious, has severe side effects with unwanted influences on the process itself or is simply not possible as markers of a specific process are not known.

To address the reviewer's criticism, we now show that pseudotime-based algorithms are not capable of reconstructing cell cycle based on image data or classical image features. The pseudotime method by Gut *et al.*, Nat. Meth. (2015) can only solve the problem when elaborate experiments on measuring known markers are carried out. This is now discussed in detail in the first section of the supplemental notes: "Pseudotime-based reconstruction of cell cycle".

We further address the reviewer's criticism by illustrating the relevance of our approach in adding an example studying disease progression. In this example, labelling information is only available by diagnoses into "healthy", "mildly diseased", "medium diseased" and "severely diseased" through trained humans. The results parallel what we observe for the cell cycle. The details are discussed in the results section after presenting cell cycle reconstruction ("Reconstructing disease progression") and in the discussion section of the main text.

Finally, we note that in many cases one might not even be aware that a few categorical labels for a dataset at hand can be interpreted as "stages of a process", for example, when one has two subtypes of a disease and it is not known whether one of these subtypes is "just a more severe form" of the other (there is a transition through the "less severe form") or whether the subtypes are similarly severe. Our method allows to reveal such scenarios: the former case would lead to a long-stretched cylinder as in the cell-cycle example and the latter case would yield to a "branching" when moving away from the healthy state. Regressing with a deep neural network or another machine learning technique, by contrast, would require the prior knowledge of the order of the labels. Also, in our experience, even if the ordering is known, regressing on qualitative labels does neither lead to an improved prediction accuracy nor to improved neural network features.

We rewrote several parts of the paper to make all of these points clearer. See the file with marked changes.

Detailed comments

** Figure 3a presents a 3D visualization of the cell "cycle", but this seems very linear. I wonder if this is an artifact of how the deep net was trained. Shouldn't we expect in theory that a cyclic structure would be the ideal lower-dimensional mapping of a cyclic event ?*

Figure 3a presents the 3D visualization of the feature representation of the neural network trained on images such as those shown in Figure 2. As in these images, the Telo phase contains images of cells that have almost finished cell cycle and resemble images of two cells, these images are highly dissimilar compared the G1 phase, whose images only show a single cell. With this type of data, a cyclic representation cannot be learned.

** propodium iodine -> propidium iodine*

We corrected the typo.

** the provided movie contained an error and did not play*

We are sorry to read that. Everything plays correctly on our side and in the link in the paper: <https://drive.google.com/file/d/0B2jSLlxOkxh1akhvcDdBbUUzeVE/view> In case the former upload was broken, we will be happy to work with the publisher to ensure this data can be accessed.

Reviewer 2

The manuscript "Deep learning for imaging flow cytometry allows reconstructing cell cycle progression" by Eulenberg et al. describes a convolutional neural network combined with non-linear dimensionality reduction to analyze imaging flow cytometry data. As a proof of principle the authors apply the algorithms to images of Jurkat cells which were imaged using bright field and two cell cycle markers by imaging flow cytometry. Their computational approach then reconstructed, without any supervision, a continuous representation of the cells along the cell cycle trajectory. The manuscript is well written and the figures illustrate the analyses in a clear and concise manner. While the deep learning field is currently exploding and lots and lots of novel applications are emerging, there is still a lot to be done in the field of biological data analysis. As such the presented approach contains enough novelty to justify publication in Nature Communications. Also the algorithm will be useful to the scientific community. Before acceptance for publication, however, the authors need to address several points.

We thank the reviewer for supporting publication of the manuscript. In the following, we thoroughly address her/his detailed comments.

1. The authors adapt the Inception architecture and describe in their text the detailed structure of the network. Yet the motivation for the actual architecture is largely absent. Why 13 modules, why 326 feature maps with size 8x8 etc.? Please explain and ideally describe why these are the optimal numbers.

We added a discussion of this point to the supplement. The main response though is that the specific choices are a matter of experience and of trial and error. We provide these parameters for others as a starting point to reproduce our and similar results.

2. The algorithm was presented using imaging flow cytometry data. While their results are impressive, imaging mass cytometry is not too common. The manuscript would greatly increase in impact if the authors showed that their algorithm provides similar results using high content fluorescence imaging data. E.g. the authors could analyze the data from Gut et al. Nature Methods 2015. They already discuss this publication and in addition could also compare the performance of their algorithm directly to an existing method.

We now compare our results with the results of Gut *et al.*, Nat. Meth. (2015) in the supplement. We explain that the approach of these authors is only suitable if one has detailed information about markers that uniquely define a process of interest. Our method does much more in that it does not require knowledge of such markers or experiments that measure them. We could not process the data of Gut *et al.* (2015) as the main step in their pipeline was the segmentation of whole fluorescence-microscopy images with a classical machine-learning algorithm and we were not able to extract the result of this segmentation separately in their pipeline. We speculate that, as an input for our pipeline, we would require a high-quality deep learning based segmentation to generate single-cell images that do not have too many artefacts of the segmentation. The latter is beyond the scope of the present paper. To nevertheless show that our workflow generalizes to other data types and processes, we added an additional example with a completely different image type dealing with disease progression. This provides strong evidence that our results are not specific to imaging flow cytometry but generalize to any kind of data and process.

3. It is also not clear for which data DeepFlow recovers cellular continua. Is the presented example a "coincidence" or will DeepFlow always find a continuum given continuous data? Some synthetic data might be helpful here.

The result is not a coincidence but holds true for any discretely-labelled continuous data, if there is a sufficient degree of overlap between categories. Only then, the relation among stages can be learned by the network, as we now more elaborately discuss in the discussion section of the main text. The new example that deals with learning a disease progression further illustrates this point.

4. On page 6, I guess the reference goes to figure 4, not 6?

We corrected the wrong reference.

REVIEWERS' COMMENTS:

Reviewer #1 (Remarks to the Author):

I appreciate the additional efforts done by the authors; they provide important new insights into why the proposed deep learning method is better at identifying the several stages of cell cycle progression.

The comparison with the pseudotemporal ordering methods learns that the classically extracted features are apparently not enough to highlight the gradual pattern in a fully unsupervised way, and that supervision (in the form of the class labels fed into the deep learning approach) is necessary to extract a good feature representation that highlights the gradual transitions. On the other hand it is quite surprising to see that the DPT algorithm seems to suffer extensively from the introduction of additional features (Figure S1, c and d), as these methods have been shown to work well on single-cell transcriptomics data, which typically contain a lot of noisy features as well.

I agree that, when keeping the Telo phase images as they are (i.e. containing two cells) this would result in a linear pattern, but one could segment out the single images and then see if a "cyclic" pattern could be found. The authors did not clearly respond whether such a cyclic pattern could be found at all using their DL approach, or whether the last layer in the network (Softmax) would automatically favour "linear" patterns.

When introducing the novel case study of disease progression, the authors mention it may not be known if some form is a "more severe form" than another one. However, looking at the network architecture in Figure 1, it seems like the phases in the classification layer are ordered based on the prior knowledge (e.g. G1 precedes S, and so on), and this ordering is used by the algorithm. If a certain disease would have three stages, but one would not know their ordering, would the proposed algorithm then still be able to recover the true gradual ordering ?

Reviewer #2 (Remarks to the Author):

The authors have addressed all concerns of the reviewers.

Reviewer #1:

I appreciate the additional efforts done by the authors; they provide important new insights into why the proposed deep learning method is better at identifying the several stages of cell cycle progression.

The comparison with the pseudotemporal ordering methods learns that the classically extracted features are apparently not enough to highlight the gradual pattern in a fully unsupervised way, and that supervision (in the form of the class labels fed into the deep learning approach) is necessary to extract a good feature representation that highlights the gradual transitions.

On the other hand it is quite surprising to see that the DPT algorithm seems to suffer extensively from the introduction of additional features (Figure S1, c and d), as these methods have been shown to work well on single-cell transcriptomics data, which typically contain a lot of noisy features as well.

Indeed, single-cell transcriptomics data contains a lot of “noisy features” but is still much more information-rich than classically extracted image features. Also, in order to make pseudotime and clustering methods work reliably, the whole community invests a lot in preprocessing of single-cell transcriptomic data. Only highly-variable genes are fed into the pseudotime algorithms. These still contain a large portion of technical noise, but, thinking about the data as originating from time-series, the running averages constitute means that correlate with observed phenotype progression. This is not the case when dealing with classical image features: many of these features do not correlate with an observed phenotype of interest; overall “intensity” might be irrelevant whereas “circularity” might be relevant. One could now strive to extract highly-variable image features etc. to achieve similar results than in single-cell transcriptomics. Nonetheless, I wouldn't expect good results for this, for what I mentioned above: marginal gene-expressions are very often, very strong features (in the sense that they correlate with the phenotype) but classically extracted image features are not.

While for gene expression data, neural networks still need to be shown to excel, there is no debate about this for (spatially-correlated) image data. Using an end-to-end deep learning solution for the latter, we circumvent the burden of manual preprocessing, parameter tuning and other cumbersome steps that are necessary in the analysis of single-cell transcriptomic data.

I agree that, when keeping the Telo phase images as they are (i.e. containing two cells) this would result in a linear pattern, but one could segment out the single images and then see if a "cyclic" pattern

could be found. The authors did not clearly respond whether such a cyclic pattern could be found at all using their DL approach, or whether the last layer in the network (Softmax) would automatically favour "linear" patterns.

The Telophase cell is still one cell but has a bi-lobular appearance as it undergoes cytokinesis. It has two nuclear poles and is still one cell as opposed to the possibility to confuse it with two cells stuck together. Segmentation is not meaningfully possible during the largest part of this stage, only at the end, when the cell splits, it would be meaningful. This very last stage would have to be extremely densely sampled in order for this to work. To try this out is beyond the scope of the paper, but we fear that the bias introduced by imperfections in this challenging segmentation task, which involves "not splitting in the Telophase too early", would destroy the whole result.

Regarding the general possibility of learning a cycle geometry in the last layer: Sorry for not having addressed this, I didn't realize that this was the aim of the question. The answer is that, yes, this is possible, even with a one-layer (linear model) in two dimensions. See the following plot, which is from Figure 4.5 of Murphy's Machine Learning book.

This is for three classes that are arranged in a "circular" way (the thick lines depict the decision boundaries). You can easily imagine more classes to fill in the "gaps" between the other classes so that a veritable "circle" emerges. This picture directly translates to the high-dimensional activation space of the last layer of a neural network. [Rather than thinking in terms of decision boundaries, one can also think of the class-specific "directions" which arise as the rows of the weight matrix in the Softmax function. For a circular geometry, these directions need to form a star. Explicit expressions for them can be found as the "betas" in Eq. (4.38) in Murphy's Machine Learning Book or in the introduction here (Wikipedia).]

When introducing the novel case study of disease progression, the authors mention it may not be known if some form is a "more severe form" than another one. However, looking at the network architecture in Figure 1, it seems like the phases in the classification layer are ordered based on the prior knowledge (e.g. G1 precedes S, and so on), and this ordering is used by the algorithm. If a certain disease would have three stages, but one would not know their ordering, would the proposed algorithm then still be able to recover the true gradual ordering ?

We are sorry that we might not have been clear enough. It is a huge advantage of our method over regression techniques that we do *not* need to know an "ordering of categories". The deep learning analysis *reveals* the order only if it is "present" in the data (in case an ordering is "present" in the data, it's last-layer activation space representation will not cluster in an unstructured fashion as it does when training a neural net on MNIST, for instance).

Thank you for your thorough criticism! It helped a lot to improve the paper!

Reviewer #2:

The authors have addressed all concerns of the reviewers.

Thank you!